# Change in Disability Associated with Psychological Distress among Internally Displaced Persons in Central Sudan

**DOI:** 10.3390/ijerph19095347

**Published:** 2022-04-28

**Authors:** Zeinat Sanhori, Lars Lien, Edvard Hauff, Touraj Ayazi, Ibrahimu Mdala, Arne H. Eide

**Affiliations:** 1Federal Ministry of Health, Khartoum P.O. Box 303, Sudan; 2Division of Mental Health and Addiction, Institute of Clinical Medicine, University of Oslo, N-0372 Oslo, Norway; edvard.hauff@medisin.uio.no (E.H.); touaya@vestreviken.no (T.A.); 3Norwegian National Advisory Unit on Concurrent Substance Abuse and Mental Health Disorders, Innlandet Hospital Trust, N-2381 Brummundal, Norway; lars.lien@sykehuset-innlandet.no; 4Department of Public Health, Innlandet University College, N-2406 Elverum, Norway; 5Unit for Education and Professional Development, Oslo University Hospital, N-0372 Oslo, Norway; 6Department of General Practice, Institute of Health and Society, University of Oslo, N-0372 Oslo, Norway; ibrahimu.mdala@medisin.uio.no; 7Department of Health Research, SINTEF Digital, Forskningsveien 1, N-0314 Oslo, Norway

**Keywords:** disability, distress, IDPs, central Sudan

## Abstract

Individuals with disabilities are particularly vulnerable in conflict settings, and a high rate of psychopathology is well documented among persons with disabilities. The objective of this study was to explore the change in disability prevalence among IDPs in two settlement areas in central Sudan and the association between disability and psychological distress. In this one-year follow-up study, 1549 IDPs were interviewed twice using the General Health Questionnaire (GHQ) to investigate emotional distress. Disability was measured using the Washington Group Short Set. Households were randomly selected using the community health center as the starting point. All household members above eighteen years of age in the sampled households were interviewed. There is an increase in disability prevalence among internally displaced persons over time, associated with rural residency and poverty, low education, unemployment, IDP status, originating from western Sudan, young age, male gender, and being married. Disability was further found to be associated with psychological distress. Disability among displaced persons should be considered as a risk factor for increase in psychopathological disorders and is closely related to poverty. The study is limited to individual-level data and does not incorporate relevant environmental variables that may have influenced the changes in disability rates.

## 1. Introduction

In order to reach the United Nation’s Sustainable Development Goals (SDGs) [1], it is of high importance to generate knowledge about particularly vulnerable populations in order to develop and implement interventions that reduce exclusion and suffering [2]. There are around 41.3 million internally displaced persons (IDPs) worldwide, most of whom live in low-income countries [3], reporting high rates of psychopathology [4,5]. While persons living with disability are over-represented among poor and vulnerable groups, it is anticipated that disability prevalence among IDPs exceeds the global estimate of 15% in any population [2]. Data and relevant research on disability in low-income countries is, however, limited [6], and high-quality internationally comparable disability data is often not available [7,8]. There are large variations in disability prevalence rates globally, caused by several factors such as differing definitions of disability, different methodologies of data collection, and variation in the quality of study design [6]. In the current study, we endeavor to estimate the prevalence of disability among IDPs in two IDP settlements in Sudan and to investigate the association between disability and psychological distress.

Individuals with disabilities are particularly vulnerable in conflict settings [9] and face many additional difficulties before, during, and after displacement. This can be due to lack of basic services and high barriers to access existing services, care, and protection. Consequences of impairments due to changes in the habitat, stigma, and negative attitudes may further aggravate deprivation of resources [10]. According to the United Nation’s Convention on the Rights of Disabled People (Article 11) (CRPD), people with disabilities need to be considered as a particularly vulnerable group during disaster and displacement, so that their specific needs are taken into account [11]. Armed conflicts, and especially civil wars, lead to increased number of people with disabilities and aggravate consequences of disability further [12]. Armed conflicts have been ongoing ever since Sudan’s independence from Britain in 1956. The conflict was rooted in economic, cultural, and religious disparities [13], and continued until 2005, when southern Sudan separated and established an independent state. Sudan’s civil war is one of the civil wars with the longest duration in the world, with truce being in place only between the first and the second civil war (1973–1983) in the past 60 years. The second civil war included other conflicts in Sudan, most notably Darfur, South Kordofan, and Blue Nile [14,15]. As a result, nearly two million people have died and more than 4 million have been displaced [16,17]. Limited data exist on the impact of long-standing conflict situations leading to displacement on disability and the relationship between disability and mental health. In this study, we therefore aim to study changes in disability rates among displaced individuals over one year and how this is related to psychological distress.

## 2. Materials and Methods

Design and setting. This is a one-year follow-up community-based study, conducted in 2011 among internally displaced adults living in two settlement areas in central Sudan. The two study areas were randomly selected in the first phase of the study. Mayo, in Khartoum state, represents an urban setting, and Moby, a rural area in the south of Gezira state, about 25 km from the state capital, Wad-Madani.

The populations of the two study areas migrated from different parts of Sudan from the outbreak of the civil war in 1983 and during the 1983, 1984, and 1985 droughts and famines that affected the Darfur and Kordofan regions. The population is from different ethnic groups and speaks different mother tongues, but Arabic is the most common language. They live in unsanitary and overpopulated areas that lack essential services such as safe water, electricity, and health services. There is a rural hospital near Mayo and a health center for the Mobi area.

Participants. The target population included all adults aged 18 and above living in the study areas. Informed consent was obtained from all subjects participating in the study. We excluded persons who declined to give informed consent. Phase 1 of the study was carried out during October 2011 and included 1876 adult IDPs. At one-year follow-up, 1549 (82.6%) of the same persons were interviewed for a second time.

Data collection. The data were collected by 20 well-trained clinical psychologists who were equally divided into two groups (for Gezira and Khartoum) with an equal gender representation. All members of the research team underwent a one-week intensive training program in research interview techniques to enhance their ability to properly utilize the research instruments and appropriately approach the community. They also conducted a pilot testing of the interview protocol. A research team from the University of Oslo and the University of Khartoum supervised them. Community guides were involved to facilitate a positive community response. Each data collection team consisted of four members—two interviewers and two community guides. All household members in the sampled households who were above eighteen years of age were interviewed.

Measurements. The questionnaire comprised sociodemographic variables, including location, sex, age, marital status, family size, education, employment, household income, and length of displacement.

Disability was measured using the Washington Group Short Set on Disability, which is an activity-based measure derived from the International Classification of Functioning, Disability and Health (ICF) [2]. The Short Set has been used in many countries across the world including Zambia [18], Sudan [19], and South Sudan [20]. It covers the following six activity domains: hearing, seeing, walking, cognition, self-care, and communication, via the following questions.

Do you have difficulty seeing even if wearing glasses?Do you have difficulty hearing even if using a hearing aid?Do you have difficulty walking or climbing steps?Do you have difficulty remembering or concentrating?Do you have difficulty with self-care, such as washing all over or dressing?Using your usual (customary) language, do you have difficulty communicating?

Answer categories were “no difficulty”, “some difficulty”, “a lot of difficulty”, and “inability” to perform the activity. The scores were used to calculate three different levels, and thus three different disability prevalence estimates [20]:

Mild to severe disability: if any difficulty in at least one of the six domains.Moderate to severe disability: if a lot of difficulty or unable to perform in at least one of the six domains.Severe disability: if unable to perform in at least one of the six domains.

Mild to severe disability represents a broad definition of disability, and severe disability is the most restrictive. Disability status as dependent variable was dichotomized according to these three cut offs.

The General Health Questionnaire (GHQ) [21] is a measure of psychological distress and has been used in various settings and cultures, including Sudan [22]. Its validity and reliability have been demonstrated extensively. The GHQ-28 is used to detect mental health problems in the general population. It assesses the respondent’s current state and asks if that differs from his or her usual state. The scoring on the 28 questions was (0–0–1–1). For instance, if the participant answered that he or she had been getting edgy and bad tempered “not at all” or no “more than usual”, the item was scored 0, whereas if response was, rather, “more than usual” or “much more than usual”, the item was scored 1. A cut-off of 5 or higher was used to indicate psychological distress. The internal reliability using Cronbach’s alpha was found to be 0.88.

A threshold of 200 SD was used after consultation with the Federal Ministry of Health to identify the poorest segment of the population.

Statistical methods. Descriptive statistics in the form of frequencies and proportions were used to describe the sociodemographic characteristics of the participants at baseline, and the normality assumptions for data on age and family size were assessed and rejected using the Kolmogorov–Smirnov test. The data on age and family size were described by medians and interquartile range, and differences between subjects from Khartoum and Gezira were determined using the Mann–Whitney U-test.

Chi-square analysis was performed to examine possible differences in disability based on three levels as well as to examine the association between the length of displacement and disability. Associations between categorical variables at baseline were established from chi-square tests of associations. We obtained prevalence and relative risk (RR) estimates for the different functional disabilities between phase I and phase II. The RR indicates how large the prevalence in phase II was relative to phase I. If the RR is less than 1, then the prevalence of disability is lower in phase II relative to phase I, and if it is greater than 1, then the prevalence of disability is higher in phase II relative to phase I. Data on disability, obtained in phase I and phase II, were nested or clustered within each participant. In this case, it is reasonable to assume that observations within individuals were not independent but, rather, were correlated. This renders inappropriate regression models such as the traditional binary logistic, which assumes independence of observations. In order to account for data nesting/clustering, we fitted a generalized estimating equation (GEE) logistic regression model to investigate the association between distress and disability. The model was adjusted for age, gender, marital status, education, employment status, household income, and place of origin. We also carried out a sensitivity analysis of the missing data using multiple imputations. Results of model testing based on multiple imputations were not significantly different from model-based data without multiple imputations. Hence, we report the results based on data without multiple imputations. All statistical analyses were performed using StataSE 14, and the level of significance was set at 5%.

## 3. Results

Table 1 shows the sociodemographic characteristics of the sample by study area; Mayo in the northern part of Khartoum with a total of 849 respondents, representing 54.8% of the total respondents, and the rural area of Mobi in Gezira with 700 (45.2%) respondents. The overall median age was 28 years and the majority were women (58.0%). Most of the respondents were married (68.6%) and were originally from western Sudan (50.7%). About 13% had permanent jobs, 23% temporary jobs, and more than half (64%) were unemployed. Economic status varied from extremely poor to poor, and 46.7% had an income of less than SDG 200 per month. Almost 19% had no formal education, 22% had received Islamic religious education (khalwa), 40% had attended elementary school, and 19% had university education.

Table 2 compares the distribution of the different disability levels in phase I and phase II. Chi-square test was used to examine the statistical significance in each comparison. Overall, there was an increasing trend for all three disability levels as cases of mild disability has increased from 90 (5.8%) to 130 (8.4%), cases of moderate disability have increased from 18 (1.2%) to 32 (2.1%), and cases of severe disability have increased from 20 (1.3%) to 21 (1.4%). The increase was statistically significant for the mild and moderate, but not for severe, disability.

The prevalence of disability by sociodemographic characteristics of the participants as well as the relative risks (RR) with phase I as the reference are presented in Table 3. The results showed that the overall prevalence of disability increased from 7.5% to 10.8% between phase I and phase II. This represents a significant increase of 44% (RR: 1.44 (1.14, 1.81)) of the prevalence at phase II compared to phase I. Gezira had relatively more cases of disability in both phases than Khartoum and more cases of disability were also observed among males than females in both phases. The prevalence of disability was 52% higher at phase II than phase I in Gezira (RR: 1.52 (1.11, 2.09)), whereas in Khartoum, the prevalence was 34% higher in phase II compared to phase I (RR: 1.52 (1.11, 2.09)). The analyses also showed that the prevalence of disability more than doubled from 2.32% in phase I to 5.32% in phase II among participants in the age group 18–29 years, whereas small relative increases were observed among participants ≥30 years. Disability cases increased significantly from 5.2% in phase I to 7.42% in phase II among those who were married, giving an RR of 1.42 (95% CI (1.08, 1.87)). Cases of disability increased from 3.10% to 4.52% among those with an elementary education and from 0.84% to 1.74% among those with at least secondary education, giving RR of 1.46 and 2.08, respectively. Among those who were unemployed, the prevalence of disability increased from 4.58% to 6.58% between phase I and phase II, respectively, giving an RR of 1.44 (95% CI: (1.07, 1.93)). We also observed a 39% increase in the prevalence of disability in phase II compared to phase I among those earning less than SDG 200 and a 50% increase among those earning at least SDG 200. The prevalence of disability increased from 4.20% to 5.75% among participants who were originally from the west of Sudan (Kordofan, Darfur). This represents a 37% increase in the prevalence of disability from phase I to phase II. 

We further analyzed the RR for three categories of disability in a forest plot (Figure 1). The three categories were 0 (without disabilities), 1–2 disabilities per individual, and ≥ 3 disabilities per individual. The prevalence of zero disability was significantly higher in phase I compared to phase II, whereas the prevalence of 1–2 functional disabilities increased from 5.8% in phase I to 9.0 % in phase II. 

Table 4 shows the odds ratios obtained by fitting the GEE logistic regression model to the data. The odds for distress increased significantly by 15% from Time 1 to Time 2 among the individuals with disabilities. However, there was no significant change in the odds of distress from Time 1 to Time 2 among individuals without disabilities (*p* = 0.58). Although the risk of distress at phase I was higher by 13% among the individuals with disabilities, the difference was not statistically significant (*p* = 0.18). We also observed that the risk for distress increased significantly by 32% in phase II among the individuals with disabilities compared to participants without any disabilities (*p* = 0.01).

## 4. Discussion

To our knowledge, this is the first longitudinal disability study in Sudan, and one of a few studies worldwide that addresses changes in disability within a group of displaced people and how this relates to emotional distress. The findings revealed significant increases in disability prevalence over one year. This increase could be due to a worsened economic situation and living in poverty over time on the edge of survival with abominable living condition, malnutrition, and diseases [23], or it could be a result of long stay as IDPs. Due to absence of data on disability prevalence in Sudan, we cannot determine whether the prevalence rates found in our study are higher, lower, or equivalent to the general population. However, a recent study from South Sudan using the same research tools for the same age group reported a disability rate of 13.4% [20], which is higher than the 10.8% prevalence found in the current study. The difference in prevalence could be due to South Sudan being poorer than the Republic of Sudan. In terms of infrastructure, South Sudan is one of the most economically disadvantaged countries in the world and has extremely scant health facilities [24].

The increase in disability prevalence was mostly due to an increase among young people in the 19–29 years age group. There was no significant increase in disability prevalence in the older age groups. This finding could imply that young persons are particularly vulnerable in situations of displacement, which finds support in previous studies [25].

The study revealed an increase in disability prevalence among the poor, persons with lower levels of education, and unemployed persons. These results are consistent with many previous studies, which have shown that persons with disabilities in developing countries have lower education, higher rates of unemployment, and are more likely to live in poverty than persons without disabilities [26]. Disability and poverty are intricately interlinked and seem to reinforce each other [6,20].

The more pronounced increase in disability among males, compared to females, is inconsistent with findings in developing and developed countries, indicating prevalence to be higher among women than men [23]. This may, however, also indicate that young male IDPs in particular are affected by the situation the study population find themselves in with limited possibilities to take traditional male roles in their community [27]. On the other hand, the displaced females, especially from the western part of Sudan, play a key role in the economic support of their families and local communities. They take part in traditional industrial, agricultural, and commercial business, in addition to domestic work in preparing food, bringing water, and gathering wood for fire [28]. This is possibly a reason for stronger resilience and resistance to adversity among females. The more pronounced increase in disability prevalence among the rural IDPs (Gezira), compared with urban IDPs (Khartoum), may be due to lack of opportunities in rural areas [7].

The study has also revealed a more pronounced increase in psychological distress among individuals with disabilities, compared to individuals without disabilities, again consistent with previous studies [9,20,29]. Persons with disabilities are particularly vulnerable in situations of crisis, poverty, and insecurity, and experience aggravated effects of bad hygienic conditions, lack of essential services and infrastructure, unfriendly environments, and absence of a disability support system [10,23,30]. While mental health and psychosocial support interventions have been implemented with support from international organizations among displaced people in Sudan during the past 10 years, little is known about the outcome and impact of these interventions.

Strengths and limitations. The strength of the study is its longitudinal design with a high response and follow-up rate. In addition, the random selection of study areas reduces selection bias and increases the strength of causal inferences but does not increase generalizability, especially since only two study sites in central Sudan were sampled. A weakness with the study is that it is limited to individual-level data and does not incorporate relevant environmental variables that may have influenced the changes in disability rates. Finally, the definition and operationalization of disability is complex and not without controversy. While the use of the Washington Group Short Set in surveys has increased over the years and has been endorsed by many influential organizations, such as the United Nations, we still acknowledge the limitation of a short set that may not fully capture the inherent complexity of disability and that self-reported data are prone to different forms of bias. We do, however, argue that the Short Set has been shown to be a practical, valid, and reliable tool for identifying disability in large surveys.

## 5. Conclusions

The study has shown that disability increased over time among IDPs living under poor conditions in two randomly sampled geographical locations in Sudan. Within this vulnerable population, young people and males, persons with disabilities, and rural dwellers seem to be particularly vulnerable to increased levels of disability and psychological stress. While it is of importance to evaluate the mental health and psychosocial support interventions that have taken place among internally displaced persons in Sudan, and further research on specific vulnerability factors within this population is needed, it is urgent to act on mental health problems and disability among IDPs. This is an area that mostly goes under the radar of both international and national authorities. In order to reach the SDGs and to act on the international vision of “leaving no one behind”, it is particularly important to cater for those who are most vulnerable and exposed to wars and disasters.

## Figures and Tables

**Figure 1 ijerph-19-05347-f001:**
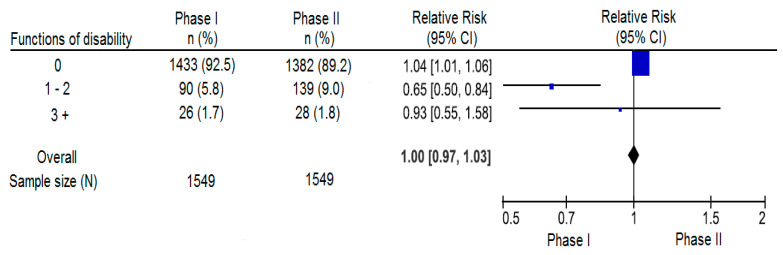
Forest plot showing the RR for three categories of number of disabilities between phase I and phase II.

**Table 1 ijerph-19-05347-t001:** Sociodemographic characteristics of 1549 IDPs at baseline (phase 1) by place of residence.

Sociodemographic	Khartoum (*n* = 849)	Gezira (*n* = 700)	Total (*n* = 1549)	*p*-Value
Age in years, median (Q1, Q3)	28.0 (22.0, 37.0)	30 (22.0, 37.0)	28.0 (22.0, 37.0)	0.31
Family size, median (Q1, Q3)	6.0 (4.0, 8.0)	6.0 (4.0, 8.0)	6.0 (4.0, 8.0)	0.78
Length of stay in years, median (Q1, Q3)	18.0 (10.0, 22.0)	18.0 (10.0, 24.0)	17.0 (10.0, 23.0)	0.92
**Gender, *n* (%)**				
Male	339 (39.9)	311 (44.4)	650 (42.0)	0.07
Female	510 (60.1)	389 (55.6)	899 (58.0)	
**Marital status, *n* (%)**				
Single	241 (28.4)	245 (35.0)	486 (31.4)	0.01
Married	608 (71.6)	455 (65.0)	1063 (68.6)	0.01
**Level of education, *n* (%)**				
Illiterate	139 (16.4)	153 (21.9)	292 (18.9)	0.01
Khalwa	220 (25.9)	127 (18.1)	347 (22.4)	0.01
Elementary	317 (37.3)	304 (43.4)	621 (40.1)	0.01
Secondary and above	173 (20.4)	116 (16.6)	289 (18.7)	0.06
**Employment status, *n* (%)**				
Unemployed	569 (67.0)	424 (60.6)	993 (64.1)	0.01
Temporary	173 (20.4)	175 (25.0)	348 (22.5)	0.03
Permanent	107 (12.6)	101 (14.4)	208 (13.4)	0.30
**Household income, *n* (%)**				
Less than SDG 200/per month	376 (44.3)	348 (49.7)	724 (46.7)	0.03
More than SDG 200/per month	473 (55.7)	352 (50.3)	825 (53.3)	0.03
**Place of origin, *n* (%)**				
North (Northern States, River Nile)	17 (2.0)	15 (2.1)	32 (2.1)	0.89
South Sudan	72 (8.5)	111 (15.9)	183 (11.8)	0.01
East (Red Sea, Kassala, Gadarif)	59 (6.9)	44 (6.3)	103 (6.6)	0.64
West (Kordofan, Darfur)	456 (53.7)	330 (47.1)	786 (50.7)	0.01
Middle (Khartoum, Gezira, Sennar, Damazeen)	245 (28.9)	200 (28.6)	445 (28.7)	0.90
**Reason for forced migration**				
War	816 (96.1)	635 (90.7)	1451 (93.7)	<0.01
Famine and drought	33 (3,9)	65 (9.3)	98 (6.3)	<0.01

**Table 2 ijerph-19-05347-t002:** Prevalence of disability based on three levels, stratified by phases.

	Time 1 *n* (%)	Time 2 *n* (%)	*p*-Value
Mild-to-severe disability: “some difficulty” in at least one of the six domains	90 (5.8)	130 (8.4)	0.04
“Moderate-to-severe disability”: “a lot of difficulty” or “unable to perform” in at least one of the six domains	18 (1.2)	32 (2.1)	0.01
Severe disability”: “unable to perform” in at least one of the six domains	20 (1.3)	21 (1.4)	0.84

**Table 3 ijerph-19-05347-t003:** The prevalence and risk (RR) of disability by sociodemographic factors with phase 1 as reference.

	Prevalence of Disability		RR (95% CI)	*p*-Value
	Phase I (*n* = 1549)	Phase II (*n* = 1549)		
Overall	7.5	10.8	1.44 (1.14, 1.81)	<0.01
**Place**				
Khartoum	3.55	4.78	1.34 (0.96, 1.89)	0.09
Gezira	3.94	6.00	1.52 (1.11, 2.09)	0.01
**Gender**				
Female	2.84	4.26	1.50 (1.03, 2.18)	0.03
Male	4.65	6.52	1.40 (1.05, 1.88)	0.02
**Age groups**				
18–29	2.32	5.23	2.25 (1.53, 3.31)	<0.01
30–39	2.91	3.03	1.04 (0.70, 1.56)	0.83
+40	2.26	2.52	1.11 (0.71, 1.75)	0.64
**Marital Status**				
Single	2.26	3.36	1.49 (0.97, 2.27)	0.06
Married	5.23	7.42	1.42 (1.08, 1.87)	0.01
**Education level**				
Illiterate	2.07	2.52	1.22 (0.77, 1.93)	0.40
Khalwa	1.48	2.00	1.35 (0.79, 2.30)	0.27
Elementary	3.10	4.52	1.46 (1.02, 2.09)	0.04
Secondary and above	0.84	1.74	2.08 (1.08, 4.01)	0.03
**Employed status**				
Unemployed	4.58	6.58	1.44 (1.07, 1.93)	0.02
Temporal	1.87	2.65	1.41 (0.88, 2.26)	0.15
Permanent	1.03	1.55	1.50 (0.80, 2.81)	0.20
**Household Income**				
Less than SDG 200	4.00	5.55	1.39 (1.01, 1.91)	0.04
SDG 200+	3.49	5.23	1.50 (1.07, 2.10)	0.02
**Original Place**				
North (Northern States, River Nile)	0.06	0.00	-	-
South Sudan	1.10	1.74	1.59 (0.87, 2.90)	0.13
East (Red Sea, Kasala, Gadarif)	0.77	0.84	1.18 (0.53, 2.63)	0.84
West (Kordofan, Darfur)	4.20	5.75	1.37 (1.00, 1.87)	0.05
Middle (Khartoum, Gezira, White Nile, Sennar)	1.48	2.45	1.65 (0.99, 2.76)	0.06

**Table 4 ijerph-19-05347-t004:** Odds ratios and 95% CIs obtained from a GEE logistic regression model showing the changes in distress between disabled and non-disabled.

	Time 1		Time 2	
	OR (95% CI)	*p*-Value	OR (95% CI)	*p*-Value
* Changes of distress within the group (ref: phase I)				
Disabled	1		1.15 (1.00, 1.34)	0.05
Not-disabled	1		0.99 (0.95, 1.03)	0.58
** Changes between the groups (ref: not-disabled)				
Disabled	1.13 (0.94, 1.35)	0.18	1.32 (1.07, 1.62)	0.01

Changes in distress among the disabled and non-disabled in phase II relative to phase I. * Changes in distress between the disabled and non-disabled (referent group) in phase I and phase II. ** Model adjusted for sociodemographic factors.

## Data Availability

Data are available upon request. The dataset will be archived in a public data repository.

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
