# Peer review of "Change in Disability Associated with Psychological Distress among Internally Displaced Persons in Central Sudan"

_ijerph, 2022, doi:10.3390/ijerph19095347_

Round 1

Reviewer 1 Report

Dear Authors,

please, find my mainly positive comments in the file attached.

Sincerely, your reviewer

Reviewer 2 Report

In my opinion, this paper provides an original and interesting insight on prevalence of disability among internally displaced persons. Its findings are valuable and stress the importance of acting on disability conditions among IDP´s.

My main concern is, however, the method used to measure disability. In my opinion, to establish that a person has a disability and to measure its level only on the basis of the answer that the person itself gives to six very simple questions is very inaccurate and might distort the results of the research. Can we be sure, for example, that a person who says that he or she has some difficulty climbing steps is a person with a disability? The same could be said concerning the other five questions. Apart from the fact that the answers can be very subjective -which means that, in presence of the same physical or mental impairment, two persons might have a different perception of difficulties- the set of questions proposed is very reduced, and does not consider, for example, difficulties in learning or in social interactions. 

Of course, I acknowledge that my objection is not to the paper itself, but to the method of measurement used by the authors. But I think that, at least, the reasons to choose this method should be specified, and the weaknesses and posible inaccuracies of such a measurement should be pointed out.

I would also like to suggest the authors to do some reference to the Convention on the Rights of Persons with Disabilities. This is today the main normative framework on disability policies, which in my opinion should be taken into account in any research on the field, Moreover, it deals specifically with the issue of the present paper, in article 11, which refers to "protection and safety of persons with disabilities in situations of risk, including situations of armed conflict, humanitarian emergencies and the occurrence of natural disasters".

Finally, although in general English language and style are fine, I suggest the authors to do a new revision. For example, in lines 53-54 the sentence "People with disabilities need to be considered as a particularly vulnerable group during disaster and displacement in order for their particular needs to be taken into account" could perhaps be changed by "People with disabilities need to be considered as a particularly vulnerable group during disaster and displacement, so that their particular needs are taken into account".
